# Fine Tuning of Calcium Constitutive Entry by Optogenetically-Controlled Membrane Polarization: Impact on Cell Migration

**DOI:** 10.3390/cells9071684

**Published:** 2020-07-13

**Authors:** Charles-Albert Chapotte-Baldacci, Guénaëlle Lizot, Cyrielle Jajkiewicz, Manuella Lévêque, Aubin Penna, Christophe Magaud, Vincent Thoreau, Patrick Bois, Stéphane Sebille, Aurélien Chatelier

**Affiliations:** 1Laboratoire Signalisation et Transports Ioniques Membranaires (STIM), CNRS ERL 7003—EA 7349, Université de Poitiers, Pôle Biologie Santé, Bâtiment B36, 1 rue Georges Bonnet, TSA 51106, CEDEX 9, 86073 Poitiers, France; charles.albert.chapotte.baldacci@univ-poitiers.fr (C.-A.C.-B.); guenaelle.lizot@univ-poitiers.fr (G.L.); cyrielle.jajkiewicz@etu.univ-poitiers.fr (C.J.); manuella.leveque@univ-poitiers.fr (M.L.); aubin.penna@univ-poitiers.fr (A.P.); christophe.magaud@univ-poitiers.fr (C.M.); patrick.bois@univ-poitiers.fr (P.B.); stephane.sebille@univ-poitiers.fr (S.S.); 2Dermatology Department, University Hospital of Poitiers, 86000 Poitiers, France; 3Neurovascular Unit and Cognitive Disorders (NEUVACOD), EA 3808, Université de Poitiers, Pôle Biologie Santé, Bâtiment B36, 1 rue Georges Bonnet, TSA 51106, CEDEX 9, 86073 Poitiers, France; vincent.thoreau@univ-poitiers.fr

**Keywords:** non-excitable cell, halorhodopsin, TRP channels, constitutive calcium entry, TRPV2, migration

## Abstract

Anomalies in constitutive calcium entry (CCE) have been commonly attributed to cell dysfunction in pathological conditions such as cancer. Calcium influxes of this type rely on channels, such as transient receptor potential (TRP) channels, to be constitutively opened and strongly depend on membrane potential and a calcium driving force. We developed an optogenetic approach based on the expression of the halorhodopsin chloride pump to study CCE in non-excitable cells. Using C2C12 cells, we found that halorhodopsin can be used to achieve a finely tuned control of membrane polarization. Escalating the membrane polarization by incremental changes in light led to a concomitant increase in CCE through transient receptor potential vanilloid 2 (TRPV2) channels. Moreover, light-induced calcium entry through TRPV2 channels promoted cell migration. Our study shows for the first time that by modulating CCE and related physiological responses, such as cell motility, halorhodopsin serves as a potentially powerful tool that could open new avenues for the study of CCE and associated cellular behaviors.

## 1. Introduction

The tight control of intracellular calcium homeostasis is a key determinant of normal cell function and survival. At any time and even in resting cells, this control involves a finely tuned balance between intracellular calcium storage, calcium influx and calcium extrusion across the plasma membrane. Calcium influx can be achieved via several classes of ion channels depending on the cell type; these include voltage-gated calcium channels, ligand-gated ion channels and channels of the transient receptor potential (TRP)/Orai families. The different types of channels facilitate calcium movement across the plasma membrane as a function of the electrochemical gradient, thereby giving rise to an increase in cytosolic-free calcium. The finely tuned control of these entries is therefore essential for the differential modulation of signaling pathways involved in specific cellular processes such as proliferation, cell death, gene transcription, cell migration, exocytosis or contraction [1].

The cytosolic calcium concentration under basal conditions achieves a steady-state equilibrium whereby calcium influx through constitutively active channels is balanced by efflux across the plasma membrane. Such constitutive calcium entry (CCE) through TRP channels has been widely related to the homeostatic function in normal cells as well as dysfunction in pathological conditions [2,3,4,5]. This form of calcium influx is due to the fact that some types of channels can be open at rest, with the amplitude of the influx strongly dependent on the membrane potential and associated calcium driving force. Whereas TRP and CCE have been identified in several cell types [3,4,5], the resting membrane potential of these cells can vary widely depending on the excitable or non-excitable nature of the cell type under investigation. Excitable cells, such as neurons or cardiac and skeletal muscle cells, are characterized by their polarized resting membrane potentials in the −50 mV to −90 mV range and their ability to generate action potentials. In contrast, non-excitable cells have a more depolarized resting potential in the −10 mV to −30 mV range [6]. Even if these cells are considered to be non-excitable, several studies have shown, however, that they exhibit membrane potential variations characterized by hyperpolarization phases during essential cellular processes such as proliferation, migration or differentiation [7,8,9]. One hypothesis is that by increasing the calcium driving force, membrane polarization would augment CCE through TRP channels and/or Orai channels. This subject has been particularly investigated in cancer cell studies but seems to be a common mechanism in many cell types [10,11,12,13,14,15,16]. The study of CCE in physiological and pathophysiological cellular processes is currently hampered, however, by the limited number of molecular tools and experimental approaches available to specifically modulate CCE.

Most studies of CCE are based on the use of gene knockout or pharmacological tools to block or activate TRP/Orai channels [17,18]. However, cells often adapt to calcium channel gene invalidation by reorganizing their calcium homeostasome to maintain calcium homeostasis. On the other hand, pharmacological modulators are often poorly selective, particularly in relation to studies requiring the long-term incubation of cell cultures with these agents where the potential for off-target cellular side effects is considerable. Moreover, the use of such pharmacological modulators often strongly activates these channels, thereby preventing their basal constitutive activity from being studied.

Here we chose to explore an alternative strategy to study CCE by modulating the driving force for calcium via the optogenetic control of the resting membrane potential. Indeed, optogenetic tools are particularly suited for fine tuning the membrane potential with a strong spatio-temporal resolution. Since the approach was first described and developed in neural cells [19], optogenetics has spurred immense research activity, permitting important advances to be made in our understanding of neural circuitry and brain function given the possibility to optically interrogate the electrical activity of targeted neurons with high specificity and spatio-temporal resolution. Whereas use of this technique has expanded to the study of other excitable cells, such as cardiomyocytes [20,21] or skeletal muscle cells [22,23,24], it has only been used sparingly to control membrane potential in non-excitable cells. As mentioned above, non-excitable cells have a lowly polarized membrane potential that may undergo temporary hyperpolarization in response to cell activity. In this condition, the use of polarizing optogenetic tools could be of particular interest. Halorhodopsin (eNpHR) is a light-activated chloride pump that drives chloride ion entry when stimulated by 590 nm light. This entry occurs even against chloride electrochemical gradient and is therefore not dependent on intracellular chloride concentration. [25]. Thus, eNpHR light stimulation leads to a negatively charged ions entry that polarizes the cell membrane potential. This pump has been principally used to date as a neuron silencer [26,27,28,29], but never to investigate the impact of membrane polarization on non-excitable cells. Optogenetics, and particularly polarizing tools such as halorhodopsin, could therefore provide a powerful means to achieve a precise spatio-temporal control of membrane potential and CCE, as well as the associated cellular processes such as cell migration.

Since this possibility has not been tested to date, we sought to determine whether membrane polarization induced by halorhodopsin activation could enhance CCE and thereby modulate cellular behavior. To test this hypothesis, we used the mouse myoblast cell line C2C12, a non-excitable cell line expressing several types of ion channels [16,30,31] amongst which TRP channels are known to be important for their calcium signaling and migration-associated properties [32,33,34,35]. After having confirmed the capacity of halorhodopsin to polarize the membrane potential of C2C12 cells, we investigated the effect of this light-controlled polarization on CCE and demonstrated that it is possible to tightly control CCE through precise halorhodopsin light stimulation. Finally, we show that continuous light stimulation of the halorhodopsin pump enhanced CCE and led to increased cell migration. During this study, we incidentally identified transient receptor potential vanilloid 2 (TRPV2) as an important player in this CCE and highlighted a role of this channel in C2C12 cell migration. We therefore propose a role for the light-responsive halorhodopsin pump as an innovative tool to investigate CCE and to modulate cellular behavior in non-excitable cells by controlling their membrane potential.

## 2. Material and Methods

### 2.1. Cell Culture and Transfection

C2C12 mouse skeletal myoblasts, human embryonic kidney (HEK)293 cells and HEK293 cells expressing mouse TRPV2 in a stable manner were grown at 37 °C with a 5% CO_2_ humidified atmosphere in Dulbecco’s modified Eagle’s medium (DMEM, Lonza, Basel, Switzerland). The DMEM was supplemented with 10% fetal bovine serum (FBS, Biowest, Nuaillé, France) and 1% penicillin/streptomycin (P/S, Sigma-Aldrich, Lyon, France) for C2C12 cells and with 8% FBS for HEK293 cell cultures. Cells were transfected using a Lipofectamine 2000 transfection reagent (Invitrogen, ThermoFisher Scientific, Villebon sur Yvette, France) according to the manufacturer’s instructions. For co-transfection, the total DNA was adjusted to 8 µg per well in 6-well plates. C2C12 myoblasts were seeded after transfection. For patch clamp and calcium imaging experiments, the cells were seeded on Matrigel (Corning, Wiesbaden, Germany)-coated glass coverslips in 35 mm tissue culture dishes, while for migration experiments, 24-well plates and Matrigel-coated glass coverslips in 35 mm tissue culture dishes were used.

### 2.2. Gene Constructs

For the optogenetic control of C2C12 myoblasts, cells were transfected with a pCMV-eNpHR3.0-EYFP plasmid derived from pCaMKIIa-eNpHR3.0-EYFP (Addgene #26971, Watertown, MA, USA). For this construct, the CaMKIIa promoter was excised using MluI and BamHI restriction endonucleases (New England Biolabs, Evry, France) and replaced by a human cytomegalovirus (CMV) promoter excised from pcDNA™3.1/Zeo(-) (ThermoFisher, Villebon sur Yvette, France) with the same enzymes. For co-expression experiments of the dominant-negative mutant E594K of TRPV2, we used the plasmid pcDNA3-mTRPV2(E594K)-Flag obtained from Aubin Penna [36].

### 2.3. Immunoblotting

To determine the TRPV2 protein expression, 1 × 10^6^ cells were lysed in a radioimmunoprecipitation assay (RIPA) buffer (50 mM Tris/HCl pH 8, 150 mM NaCl, 5 mM EDTA, 0.05% NP-40, 1% sodium deoxycholate, 1% Triton X-100, 0.1% SDS, Sigma-Aldrich, Lyon, France) containing protease and phosphatase inhibitors (Protease Inhibitor Cocktail, Sigma-Aldrich, Lyon, France; PhosSTOP Phosphatase Inhibitor Cocktail, Roche, Mannheim, Germany). Next, 10 µg of protein samples were separated by SDS/PAGE and transferred to nitrocellulose membranes. Membranes were then subjected to western blotting using a rabbit anti-TRPV2 antibody (1:1000, ab74859, Abcam, Cambridge, UK) and a mouse anti-glyceraldehyde 3-phosphate dehydrogenase (GAPDH) antibody (1:1000, sc-32233, Santa Cruz, Heidelberg, Germany). Horseradish peroxidase-conjugated goat anti-rabbit and goat anti-mouse antibodies (1:5000, 115-035-144 and 115-035-146, Jackson ImmunoResearch, Ely, UK) were used as secondary antibodies and proteins were detected using enhanced chemiluminescence (Immobilon, Merck Millipore, Molsheim, France). Images were obtained using the GeneGnome Imager (SynGene Ozyme, Montigny-le-Bretonneux, France).

### 2.4. Immunofluorescence Staining

C2C12 myoblasts transfected with eNpHR-YFP alone or co-transfected with mTRPV2[E594K]-FLAG were fixed using 4% paraformaldehyde (PFA, Sigma-Alrich, Lyon, France) in phosphate-buffered saline (PBS, Corning, Wiesbaden, Germany) for 10 min at 4 °C and permeabilized in 0.2% PBS/Triton X-100. Blocking of non-specific sites and permeabilization were achieved by incubation with PBS containing 1% bovine serum albumin (BSA, Sigma-Aldrich, Lyon, France) and 0.1% Tween-20 (Sigma-Aldrich, Lyon, France) for 30 min at room temperature. Cells were stained using the blocking solution containing rabbit polyclonal anti-TRPV2 antibody (1:200, ACC-039, Alomone labs, Jerusalem, Israel) or rabbit polyclonal anti-FLAG antibody (1:100, F7425, Sigma-Aldrich, Lyon, France) overnight at 4 °C, followed by an incubation with the donkey anti-rabbit secondary antibody coupled with red fluorescent Alexa Fluor 555 (1:400, Molecular Probes, ThermoFisher Scientific, Villebon sur Yvette, France) for 2 h at room temperature. Finally, DAPI (4′,6-Diamidino-2-phenylindole dihydrochloride, 1:200, Sigma-Aldrich, Lyon, France) staining was performed to visualize nuclei. After washing, the glass coverslips were slide mounted in Mowiol (Sigma-Aldrich, Lyon, France). The immunolabeled samples were examined by confocal scanning microscopy using an FV-1000 system mounted on an Olympus IX81 inverted microscope (Olympus, Tokyo, Japan).

### 2.5. Reverse Transcription and Polymerase Chain Reaction (RT-PCR)

Total RNA from cultures of C2C12 myoblasts was isolated using RNABLe reagent (Eurobio, Les Ulis, France) followed by chloroform extraction and isopropanol precipitation (Sigma-Aldrich, Lyon, France). RNA integrity was evaluated by ethidium bromide (Sigma-Aldrich, Lyon, France) staining on a 1% agarose gel. Total RNA was quantified by assessing optical density at 260 and 280 nm (NanoDrop ND-100 Labtech, Thermo Scientific, Villebon-sur-Yvette, France). Complementary DNAs (cDNAs) were synthesized as follows: 10 μL of total RNA (1 µg) were added to 12 μL of reaction mixture (100 mM Tris–HCl (pH 8.3), 150 mM KCl, 6.25 mM MgCl2, 20 mM DTT, 2 mM dNTPs) (Invitrogen, ThermoFisher Scientific, Villebon sur Yvette, France) and 1.5 μg Random Primer Pd(N)6 (Invitrogen, ThermoFisher Scientific, Villebon sur Yvette, France). RNA was denatured at 65 °C for 2 min and then added to 40 U RNAse inhibitors (RNaseOUT, Invitrogen, ThermoFisher Scientific, Villebon sur Yvette, France) and 400 U M-MLV Reverse Transcriptase (Invitrogen, ThermoFisher Scientific, Villebon sur Yvette, France) to 25 μL final volume. cDNA was synthesized at 37 °C for 1 h and then 25 μL sterile water was added. The remaining enzymes were heat-deactivated (100 °C, 2 min). After the RT procedure, 10 μL of cDNA (≈150 ng) was added to 40 μL of PCR reaction mixture (22 mM Tris–HCl (pH 8.4), 55 mM KCl, 2.2 mM MgCl_2_, 277.8 µM dNTPs, 12 pmol forward and reverse primers and 1.25 U of Taq Polymerase) (Invitrogen, ThermoFisher Scientific, Villebon sur Yvette, France). Thermal cycles were performed in a PTC-100 thermal cycler (M.J. Research, Inc, Watertown, MA, USA) and consisted of exposure to 94 °C for 5 min, followed by 35 cycles at 94 °C for 30 s, 55, 57 or 60 °C for 1 min and 72 °C for 1 min. After the last cycle, samples were incubated at 72 °C for 5 min to ensure complete product extension. All primer sequences and annealing temperatures are described in Table 1. An amount of 18S mRNA was used as a housekeeping gene and the negative control consisted of PCR reactions without template addition. cDNAs extracted from hippocampal and cortical tissue were used as positive controls for TRPC4 mRNAs [37]. Amplified products were separated by electrophoresis on 2% agarose gels (containing 0.01% ethidium bromide) in Tris-Acetate-EDTA buffer (Sigma-Aldrich, Lyon, France) and visualized using a UV Transilluminator (E-box VX5, Vilber, Marne la Vallée, France).

### 2.6. Electrophysiology

Endogenous fluorescence of yellow fluorescent protein (YFP) was used to identify eNpHR-positive C2C12 myoblasts. Measurements were carried out at room temperature (≈22 °C). Patch electrodes (≈4 MΩ) were pulled from borosilicate glass capillaries (GC150T, Harvard Apparatus, Les Ulis, France) using a vertical micropipette puller (Narishige, Tokyo, Japan). The patch pipettes were filled with (mM): 10 NaCl, 130 KCl, 0.5 MgCl_2_, 2 Mg-ATP, 1 EGTA and 10 HEPES. The pH was adjusted to 7.2 using KOH. The bath solution contained (mM): 140 NaCl, 5.4 KCl, 1.8 CaCl_2_, 1.8 MgCl_2_, 11 glucose, and 10 HEPES. The pH was adjusted to 7.4 using NaOH. Experiments were performed using an Axopatch 200B amplifier with a CV 202 AU headstage (Molecular Devices, San Jose, CA, USA). Voltage and current clamp (I = 0) experiments were performed using the whole-cell configuration of the patch-clamp technique. Voltage-clamp and light pulses were generated by a personal computer equipped with an analog-digital converter (Axon Digidata 1550a, Molecular Devices, San Jose, CA, USA) using pClamp software v10.2 (Molecular Devices, San Jose, CA, USA). eNpHR currents were digitized at 5 KHz and filtered at 2 KHz. For all experiments, a 5 s interval between each stimulation was applied. The digitized currents were stored on a computer for later off-line analysis. Cell illumination was performed through the 20x objective of the microscope by using a light guide-coupled LED with a 590 nm beam (Thorlabs, Maisons-Lafitte, France). This was connected to a DC4100 controller (Thorlabs, Maisons-Lafitte, France) and driven via pClamp 10.2 (Molecular Devices, San Jose, CA, USA) to manage light pulses. Voltage-clamp experiments were performed on cells maintained at a holding potential of −15 mV and exposed to 1 s light pulses of different intensities (from 2.5 to 84.1 mW.cm^−2^). A 3 min light pulse at 16.2 mW.cm^−2^ was performed to observe voltage control stability induced by eNpHR activity. Analyses were performed using Clampfit 10.2 (Molecular Devices, San Jose, CA, USA). Light power was assessed by a photometer (Opton Laser International, Orsay, France) at the output of the 20X microscope objective.

### 2.7. Calcium Measurements

Intracellular calcium concentration ([Ca^2+^]_i_) changes were measured using the Fura-2/AM fluorescent probe (ThermoFisher Scientific, Villebon sur Yvette, France). C2C12 myoblasts expressing eNpHR-YFP were incubated with Fura-2/AM (3 μM, ThermoFisher Scientific, Villebon sur Yvette, France) for 30 min at 37 °C with 5% CO_2_. They were washed twice with Tyrode’s solution containing 130 mM NaCl, 5.4 mM KCl, 0.8 mM MgCl_2_, 1.8 mM CaCl_2_, 10 mM HEPES, 5.6 mM glucose (pH 7.4). Petri dishes were mounted in the observation chamber and cells bathed in Tyrode’s solution were imaged. Ratiometric calcium imaging was performed with an Olympus IX73 inverted microscope (Olympus, Tokyo, France). Cells were excited at 340 and 380 nm using a Lambda 421 beam combiner (Sutter Instrument, Ballancourt-Sur-Essonne, France) and the emitted signal acquired at 510 nm using an Andor Zyla 4.2 PLUS cooled sCMOD camera (Andor Technology, Oxford Instruments, Belfast, UK). Images were acquired with Metafluor software (Molecular Devices, San Jose, CA, USA). Paired images were collected every 1–2 sec for 1-10 min and analyzed with ImageJ software (NIH, Bethesda, MD, USA). Basal fluorescence was recorded for at least 30 s, after which the coverslips were used for light stimulation and treatments. Fluorescence changes were expressed as the ratio F340/F380 normalized to basal values (ΔF/F0). eNpHR-YFP-positive cells were identified by YFP fluorescence. Cell illumination was performed using a 593/40 nm LED light positioned close to the cell and controlled manually by a Ce:YAG driver (Doric Lenses, Quebec, Quebec, Canada). Light pulses (duration 30 s) of different intensities (from 2.5 to 56.1 mW.cm^−2^) were applied to cells in Tyrode’s solution in the presence or absence of calcium, and with or without preincubation for 30 min with 100 µM Tranilast (Merck Millipore, Molsheim, France). For experiments with Ca^2+^-free solution, CaCl_2_ was omitted from the medium and replaced with 0.1 mM EGTA.

### 2.8. Fura-2 Quenching Assay

eNpHR-YFP-expressing myoblasts, plated on glass coverslips, were washed with Tyrode’s solution and incubated with Fura-2/AM (3 μM) for 30 min at 37 °C and 5% CO_2_. After Fura-2/AM loading, cells were perfused with Tyrode’s solution for a few seconds and then a peristaltic pump (Gilson, Middleton, WI, USA) was used to perfuse the cells with a manganese solution (130 mM NaCl, 5.4 mM KCl, 0.8 mM MgCl_2_, 0.1 mM Mn^2+^, 10 mM HEPES, 5.6 mM glucose, pH 7.4) to quench the Fura-2 fluorescence. Fura-2-loaded cells were excited at 365 nm with a Lambda 421 beam combiner and the emitted signal was acquired at 510 nm using the same system described in the section “Calcium measurements”. The influx of Mn^2+^ through cation channels was evaluated by the quenching of Fura-2 fluorescence excited at 365 nm, i.e., at the isosbestic point (360 nm). Fluorescence variation was recorded with Metafluor software (Molecular Devices, San Jose, CA, USA). After 3 min in manganese solution, cells were illuminated with a 593/40 nm LED light at an intensity of 48 mW.cm^−2^. C2C12 myoblasts which did not express eNpHR-YFP were used as internal control. The quench rate of fluorescence intensity, expressed as percent per minute, was estimated using a linear regression analysis and the intensity obtained before Mn^2+^ perfusion was set to 100%.

### 2.9. In Vitro Migration Assay

3 × 10^3^ cells/cm^2^ were grown on 24-well plates or in 35 mm-diameter culture dishes with a bottom made from a glass coverslip coated with Matrigel. Cell migration was recorded automatically every 20 min for 15 h for cells in a culture medium maintained at 37 °C and 5% CO_2_. eNpHR-YFP-positive cell migration tracking was performed using the YFP fluorescence. Experiments using eNpHR light stimulation assays and associated controls were performed on a fast spinning disk confocal microscope (IX81-ZDC, Olympus, Tokyo, Japan) mounted on an Andor Revolution imaging system (Andor Technology, Oxford Instruments, Belfast, UK) and coupled to light stimulation at 590 nm with an optical fiber (Doric Lenses, Canada). C2C12 myoblasts were continuously stimulated at 18 mW.cm^−2^ throughout the migration assay (15 h). A second part of the cell migration experiment, which did not require light stimulation, was performed using a real-time recorder (JuLI stage, NanoEnTek, Seoul, Korea) in combination with treatments. Cell tracking was performed with ImageJ software and average velocity from multiple independent coverslips was used to calculate the sample mean ± standard error of the mean (SEM) obtained after analysis.

### 2.10. Statistical Analysis

Results were expressed as mean ± SEM. All statistical analyses were performed using PRISM software (GraphPad, San Diego, CA, USA). Kruskal–Wallis and Mann–Whitney tests were used to determine statistical significance.

## 3. Results

### 3.1. Light Activation of eNpHR Induces Membrane Polarization in C2C12 Myoblasts

To determine the impact of light-induced eNpHR activation on chloride currents and membrane potential, we expressed a recombinant eNpHR-YFP fusion protein in C2C12 myoblasts by cellular transfection of the pCMV-eNpHR3.0-YFP plasmid. The vector codes for a 590 nm wavelength-activated chloride pump coupled to the yellow fluorescent protein (YFP) to identify transfected cells (Figure 1A). Twenty-four hours after transfection, the expression, localization and functionality of eNpHR-YFP in C2C12 myoblasts were determined. We observed a strong expression of eNpHR-YFP protein located primarily at the plasma membrane, although a minor but detectable signal around intracellular sites was also observed (Figure 1B). The functionality of eNpHR in transfected C2C12 myoblasts was assessed by evaluating its ability to be activated by 1 s light pulses of escalating intensity. Light stimulations (590 nm) were applied to patch-clamped (whole-cell configuration) C2C12 myoblasts held at −15 mV. Light stimulation induced outward currents from a light power of 1.3 mW/cm^2^, producing a current of 0.2 ± 0.1 pA/pF (*n* = 29) and reaching a plateau of 5.2 ± 1.1 pA/pF at around 40.2 mW/cm^2^ (*n* = 29; Figure 1C). To examine changes in the membrane potential induced by eNpHR currents, C2C12 myoblasts were placed in the current-clamp configuration and irradiated with 1 s light pulses as before. The increase in light power induced cell polarization, with a shift of the membrane potential toward more negative values (Figure 1D). The resting membrane potential of these cells was −9.3 ± 2.3 mV in the absence of light stimulation. Membrane potential polarization commenced at a light power of 2.7 mW/cm^2^ (−15.2 ± 2.7 mV, *n* = 36) and hyperpolarized towards a plateau beginning at irradiations above 29.2 mW/cm^2^. The membrane potential continued to decrease more gradually until a rheobase of −87.8 ± 7.3 mV was reached at 84.1 mW/cm^2^ (*n* = 36). At maximum light intensity, the kinetics of membrane polarization are depicted by a time constant of 18.7 ± 2.1 ms (*n* = 36). To test whether membrane polarity could be maintained for long periods of light stimulation, light (16.2 mW/cm^2^) was applied for 180 s. The membrane potential decreased, reaching a steady-state level around −50 mV and then returning to the basal value of −10 mV once the light stimulation was switched off (Figure 1E). These results indicate that the halorhodopsin pump is a relevant tool for the fine and reversible control of membrane polarization. We therefore sought to test the impact of this pump’s activity on the maintenance of intracellular calcium homeostasis.

### 3.2. Light-Activated Membrane Polarization Induces Calcium Elevation through Constitutive Ca^2+^ Entry

Membrane polarity is a determining factor in the control of calcium influx. Indeed, membrane polarization increases the calcium driving force and could therefore magnify CCE [5]. To test this hypothesis in our C2C12 model, we performed experiments to measure changes in [Ca^2+^]_i_ that may occur during light-induced membrane polarization. We used a strategy based on the ratiometric Fura-2 calcium-sensitive dye. Conveniently, the excitation/emission wavelengths of Fura-2 do not overlap with those of YFP or eNpHR, thus permitting simultaneous Fura-2 recordings and eNpHR stimulation to be performed. Light stimulations at 590 nm led to increased [Ca^2+^]_i_ in eNpHR-transfected myoblasts, in contrast to control cells where no calcium increase was observed (Figure 2A). The lowest calcium response was obtained for light stimulations of 6 mW/cm^2^, with a plateau reached for values above 48 mW/cm^2^ (Figure 2B). Increased [Ca^2+^]_i_ was seen almost immediately from the time the light stimulation was turned on and plateaued throughout the duration of light stimulation (Figure 2C). When the light stimulation was switched off, [Ca^2+^]_i_ decreased gradually back to its basal level with a mean recovery time of 50.6 ± 2.8 s. To determine whether the calcium increase depended on intracellular or extracellular stores, C2C12 cells expressing eNpHR-YFP were perfused with Tyrode’s solution containing no calcium. No light-induced calcium elevation was observed during the perfusion of cells with this solution, which contrasted to that seen with the perfusion of Tyrode’s solution containing calcium (Figure 2C,D). Washout of the calcium-free solution with a control of Tyrode’s solution restored light-induced calcium elevations, but to a lower level. To confirm the extracellular origin of the calcium source, we conducted Mn^2+^ quenching experiments and compared the rate of Fura-2 quenching as an index of calcium entry. A significant increase in Fura-2 fluorescence quenching (Figure 2E) was observed in light-stimulated C2C12 myoblasts expressing eNpHR-YFP (23.7 ± 3.0 %/min) compared to control cells (9.6 ± 0.5 %/min) (Figure 2F). These results suggest that light-induced membrane hyperpolarization augments CCE, thereby increasing [Ca^2+^]_i_.

### 3.3. Expression of TRPV2 Channels in C2C12 Myoblasts

Polarization-dependent calcium elevation could require the activation of different types of calcium-permeable ion channels. To this end, calcium entry has been shown to occur in human skeletal myoblasts via T-type calcium channel window currents [7]. However, this hypothesis is unlikely in C2C12 myoblasts since it has been shown that these channels are not expressed in rat and mouse myoblasts [38,39]. In addition, patch clamp experiments on C2C12 myoblasts did not reveal the presence of any voltage-dependent calcium currents (15 tested cells; data not shown). A second possibility could be the potentiation of CCE through TRP channels. Indeed, the presence of transient receptor potential canonical channel 1 (TRPC1) [33], transient receptor potential melastatin channel 7 (TRPM7) and transient receptor potential vanilloid 2 (TRPV2) [35] channel activity was shown in skeletal muscle myoblasts, with these channels controlling processes such as cell migration or cellular fusion [33,40]. Of note, TRP channels have been shown to mediate CCE in different systems [5]. To determine the molecular player(s) potentially involved in C2C12 light-driven calcium entry, we used RT-PCR to detect the possible expression of different TRP channels in these cells. In this way, TRPC1, TRPV2, TRPV4 and TRPM7 transcripts were detected in our cell model, while no signal for TRPC4 was found (Figure 3A and Appendix A). Of the expressed channel transcripts, the properties of TRPV2 [14,36] render it a good candidate to be mediating light-induced calcium entry. The expression and localization of TRPV2 channels in C2C12 myoblasts were further investigated in western-blot experiments, with TRPV2 channel proteins detected at a molecular weight around 97 kDa (Figure 3B). As a control, HEK cells stably expressing mouse TRPV2 were used; these showed a specific band at 86 kDa, which was absent in control HEK cells (Figure 3B). The difference in molecular weight (97 kDa vs. 86 kDa) could be due to differences in glycosylation patterns as previously demonstrated in other cell models [41,42,43]. As it was shown that proper trafficking of TRPV2 to the plasma membrane is required for it to control CCE [41], we also performed immunocytochemistry staining. These experiments confirmed the presence of endogenous TRPV2 channels located in the plasma membrane of C2C12 myoblasts under normal growth conditions (Figure 3C).

### 3.4. Involvement of TRPV2 in Calcium Response to Optical Stimulation

We subsequently focused our attention on TRPV2 since this channel is a well-known player in CCE, is expressed in various cellular systems, and has a relatively well developed panel of pharmacological and molecular tools available to block its activity such as Tranilast [44,45,46,47] or dominant-negative mutants [36,48,49], respectively. To determine the role of TRPV2 channels in the light-induced calcium response (Figure 4), variations in Fura-2 fluorescence during light stimulation were monitored in C2C12 myoblasts expressing eNpHR-YFP and treated with either the DMSO vehicle as control or 100 µM Tranilast to block TRPV2 activity. Interestingly, exposure to Tranilast interfered with the light-induced increase in [Ca^2+^]_i_ in C2C12 myoblasts (Figure 4A), leading to a 77% reduction in the peak level of the control Ca^2+^ transient (Figure 4B). Tranilast is considered to be the most specific, commercially available TRPV2 inhibitor; however, the possibility of other molecular targets being implicated in its action cannot be ruled out. To confirm the implication of TRPV2 channels in the light stimulation response, eNpHR-expressing C2C12 myoblasts were co-transfected with a plasmid encoding a Flag-tagged TRPV2 channel bearing the E594K mutation conferring a dominant-negative activity [48]. As shown in Figure 4C, most cells overexpressing eNpHR-YFP also co-expressed TRPV2[E594K] mutant channels (Figure 4C). To examine the impact of TRPV2[E594K] expression on the light-induced calcium response, we monitored the variation in Fura-2 fluorescence during light stimulation in myoblasts expressing eNpHR-YFP alone or co-transfected with the dominant-negative mutant. A decrease of 55% in the peak amplitude of the Ca^2+^ transient (Figure 4D) was observed in cells expressing the TRPV2[E594K] mutant compared to control cells (Figure 4E). These results are indicative of a strong contribution of TRPV2 channels to the light-induced calcium entry seen in C2C12 myoblasts.

### 3.5. Optogenetic Control of TRPV2-Dependent C2C12 Myoblast Migration

We next considered whether the light-induced control of [Ca^2+^]_i_ could be used as a tool to fine-tune calcium-dependent physiological processes. Depending on the cellular context, involvement of the TRPV2 channel is thought to play a role in different physiological and pathological processes through the control of cell adhesion, proliferation, migration or invasion [5,43,50]. Here, we determined if TRPV2 regulates C2C12 myoblast migration and, if so, how this process could be modified by the light stimulation technique. Compared to control (0.412 ± 0.011 µm/min, *n* = 432) and DMSO-treated (0.426 ± 0.017 µm/min, *n* = 180) C2C12 myoblasts, basal cell migration was significantly decreased in C2C12 myoblasts expressing the TRPV2[E594K] mutant (0.318 ± 0.011 µm/min, *n* = 290, *p* < 0.0001), treated with 100 µM Tranilast (0.311 ± 0.009 µm/min, *n*= 285, *p* < 0.001), or both expressing TRPV2[E594K] and treated with 100 µM Tranilast (0.299 ± 0.017 µm/min, *n* = 89, *n* < 0.001). It should be mentioned that no significant differences were observed between control and DMSO-treated C2C12 myoblasts (Figure 5A). These results thus demonstrate that TRPV2 is involved in the migration of C2C12 myoblasts. To investigate the impact of light stimulation on the migratory behavior of TRPV2-expressing C2C12 myoblasts, we exposed cells expressing eNpHR-YFP to a continuous light stimulation (590 nm; 17 mW/cm^2^) and tracked their movement.

Light-stimulated C2C12 myoblasts migrated faster compared to unstimulated cells (Figure 5B). Indeed, whereas control cells migrated at a velocity of 0.446 ± 0.023 µm/min (*n* = 115), the light-stimulated C2C12 myoblasts moved at 0.579 ± 0.024 µm/min (*n* = 125, *p* < 0.001). This effect was inhibited when cells were co-transfected with the dominant-negative mutant of TRPV2 (0.398 ± 0.014 µm/min, *n* = 162), when cells were treated with 100 µM Tranilast (0.400 ± 0.017 µm/min, *n* = 133), or when they were concomitantly transfected with the dominant-negative mutant and treated with 100 µM Tranilast (0.339 ± 0.013 µm/min, *n* = 163). Note that this last condition decreased the migratory velocity to values significantly lower than the control value obtained in the absence of illumination (*p* < 0.001) (Figure 5C). These data thus clearly demonstrate that the optogenetic polarization of C2C12 cells via the halorhodopsin pump increases their migratory properties through a TRPV2-dependent pathway.

## 4. Discussion

Subtle but durable changes in the resting [Ca^2+^]_i_ have an impact on the cell phenotype and can be seen as signaling events per se. As CCE is an important factor determining [Ca^2+^]_i_, the former’s modulation provides an important mechanism to alter cellular physiological processes such as cellular proliferation, migration and differentiation. Depending on the cellular model, CCE has been correlated to the expression of several ion channels, among which members of the TRP channel family seem to play a central role. Studies often investigate CCE-dependent roles by using pharmacological approaches to modulate the activity of these channels and to look at the subsequent impact on cellular properties such as migration or proliferation. However, the use of such molecules does not allow for fine spatio-temporal control to be achieved over CCE. Side effects on calcium homeostasis and cellular behavior can also result from off-target effects on other pathways. Other investigations have addressed the role of CCE by modulating membrane potential via the pharmacological inhibition of ion channels such as volume-regulated anion channels (VRAC), large-conductance calcium-activated potassium channels (BKCa) or intermediate-conductance calcium-activated potassium channels (IKCa) [11,14,16]. These approaches have the advantage that they decrease the driving force for calcium entry and modulate CCE. However, they also have drawbacks in that they prevent the membrane potential from being tightly controlled and limit any spatial or temporal resolution of this parameter. Indeed, investigating the impact of CCE on cell behavior using these different strategies necessitates incubating cells with drugs that necessarily impact on all cells within the culture and prevent the targeting of specific cells or subcellular areas. The possibility to study the impact of membrane potential variations on CCE and cell behavior is also limited. In this context, our work achieves a significant technical advance by establishing an easy, accessible and tunable way to precisely control CCE by modulating the calcium driving force via activation of the halorhodopsin pump. Given the light activation properties of halorhodopsin, this new and innovative approach enables CCE to be controlled with a strong spatial and temporal resolution. This in turn opens new possibilities for the study of cellular behaviors such as cell migration associated with the modulation of calcium entry.

Optogenetics was originally developed to control membrane potential and thus stimulate or inhibit cellular excitability [26,28,51]. More specifically, the halorhodopsin pump was first used to block action potential generation by polarizing neurons with 590 nm light, an approach that required a light power density in the mW/mm^2^ order of magnitude [28]. Surprisingly, lower light power densities (in the mW/cm^2^ range) were sufficient to induce a strong membrane polarization of up to 100 mV in our C2C12 model. This discrepancy could be explained by several factors such as different halorhodopsin expression levels in different cell models, or that C2C12 cells are non-excitable and display a poorly polarized (around −10 mV) resting membrane potential. Non-excitable cells often display a small background conductance as opposed to polarized cells that harbor a background potassium conductance. In accordance with Ohm’s law, this larger basal membrane resistance should lead to a stronger impact of small currents on variations in membrane potential. As a consequence, small halorhodopsin-mediated currents would have a strong impact on membrane polarization. These differences between excitable and non-excitable cell models are important because they infer that halorhodopsin is well adapted for a use in non-excitable cells and represents a powerful tool to achieve membrane polarization.

In our study, we showed that an escalating membrane polarization could be achieved by incrementing light intensity, and that this was associated with an increase in calcium driving force, leading to a concomitant increase in calcium entry. [Ca^2+^]_i_ increased over the duration of the stimulation and decreased progressively when the light stimulation was switched off. This point is particularly interesting since it clearly shows that illumination provides a tight and reversible control over CCE at previously unattainable levels in terms of amplitude and temporal resolution. By removing the extracellular calcium, or replacing it with manganese, we were able to show that the light-stimulated increase in [Ca^2 +^]_i_ was a consequence of extracellular calcium entry into the cells via constitutively open channels. In the literature, CCE has been related to Orai and TRP channels; some of these channels are constitutively open and are associated with a range of cellular processes as well as cancer formation [5,52]. We therefore sought to characterize the molecular identity of the channels involved in CCE in our cell model. Previous studies have identified Orai [53] and TRP channels such as TRPC1 [33], TRPM7 and TRPV2 [35] as being expressed in the C2C12 cell line. All of these channel types could participate in CCE, but our results revealed a strong expression of TRPV2 channels in the plasma membrane. Since the TRPV2 inhibitor Tranilast and the negative-dominant TRPV2[E594K] inhibited an important part of the light-induced CCE, TRPV2 seems to be the main player involved in this light-induced calcium entry. Nevertheless, since a small part of calcium entry remained after the Tranilast treatment or TRPV2[E594K] expression, it is possible that other channels such as TRPC1 and Orai could also participate in light-induced CCE, but to a lesser extent. Moreover, it is not excluded that a calcium-induced calcium release (CIRC) mechanism takes part in light-induced calcium elevation. It is well described that an increase in intracellular calcium can activate CICR through ryanodine receptors (RyR) and inositol 1,4,5-trisphosphate receptors (IP3R). In our experiments (Figure 2C), the amplitude of the light-induced calcium response was lower after the perfusion of the free calcium solution compared to the first stimulation. This could be the consequence of a lower level of calcium in intracellular stock compared to the first stimulation. 

Cell migration is an important cellular function involved in numerous physiological and pathological processes [54]. It is regulated by multiple calcium-dependent pathways impacting on cytoskeletal remodeling, modulation of focal adhesion or cell contraction [55,56,57,58]. In this way, CCE-dependent resting calcium levels directly affect the motile behavior of the cell [52,59,60]. This relationship has been well described in cancer where CCE is often deregulated and associated with an increase in cell migration [52]. TRPV2 was reported to be involved in this process in many cellular models such as PC-3 [43,50] and LNCaP [61] cancer cell lines where it increases their migration properties. TRPV2’s translocation to the plasma membrane and its association with BKCa also increased MDA-MB-435s cell migration [14]. TRPV2 has previously been described in C2C12 cells [35], but its implication in cell migration and differentiation has not been reported. Our study demonstrates for the first time that this channel plays an important role in the migration of C2C12 cells.

TRPV2’s involvement in the light-induced calcium increase and its function in C2C12 migration prompted us to test whether halorhodopsin-mediated stimulations could impact cell migration properties. Our results showed that light-induced CCE increased cell migration and that this effect was strongly reduced following TRPV2 inhibition. This revealed the role of TRPV2 in myoblast migration and confirmed that halorhodopsin pump activation can be used to control CCE to modulate specific aspects of cell behavior. Hence, our study shows for the first time that halorhodopsin could serve as a powerful tool to modulate membrane polarization and thus CCE, thereby enabling the controlled manipulation of non-excitable cell physiological responses such as cell motility (Figure 6).

In conclusion, our optogenetic approach appears well adapted for the study of calcium responses induced by fast membrane polarization. It also offers the advantage of a strong spatial and temporal control of the membrane potential. In a previous study, we showed that targeted optogenetic activation of ChR2 at a subcellular level can modulate calcium homeostasis [24]. By applying a similar strategy of stimulations to small membrane areas, the spatial restriction of halorhodopsin activation could be achieved to increase local constitutive calcium entry at specific cellular sites, such as podosomes, lamellipodia or pseudopodia, where calcium entry through TRP channels interacts with cytoskeletal remodeling processes [14,60,62]. Indeed, several studies have shown that the subcellular localization of TRP channels could be important for cellular processes such as migration. As an example, TRPC6 was shown to colocalize with BKCa in podocytes, with TRPC6 possibly serving as calcium source for BKCa activation [10]. Similarly, Gambade and colleagues showed that the translocation of TRPV2 to pseudopodia induced calcium entry and increased cancer cell migration [14]. Therefore, our halorhodopsin-based approach could provide crucial information on the role of such calcium entry in specific cellular microdomains. Using the halorhodopsin optogenetic approach in non-excitable cells could also provide new opportunities to investigate the impact of different membrane polarization patterns on cell behavior in cell culture. Indeed, while non-excitable cells are known to display membrane potential oscillations during different cellular processes [8,9], the impact of these oscillations on the cell phenotype remains poorly understood. The high temporal resolution for the control of membrane polarization and associated CCE obtained with halorhodopsin-mediated stimulation should render this kind of investigation more feasible. Besides, progressive membrane polarization occurs during the initiation of the myoblast fusion process leading to an increase in intracellular calcium [7]. The approach described in our study could be helpful to study myogenesis by mimicking the progressive membrane polarization observed in the physiological condition using for example, a slow ramp of light. Finally, our approach provides an easy way to simultaneously augment and monitor CCE and could therefore provide an interesting and powerful method for the screening of pharmacological agents that modulate CCE.

## Figures and Tables

**Figure 1 cells-09-01684-f001:**
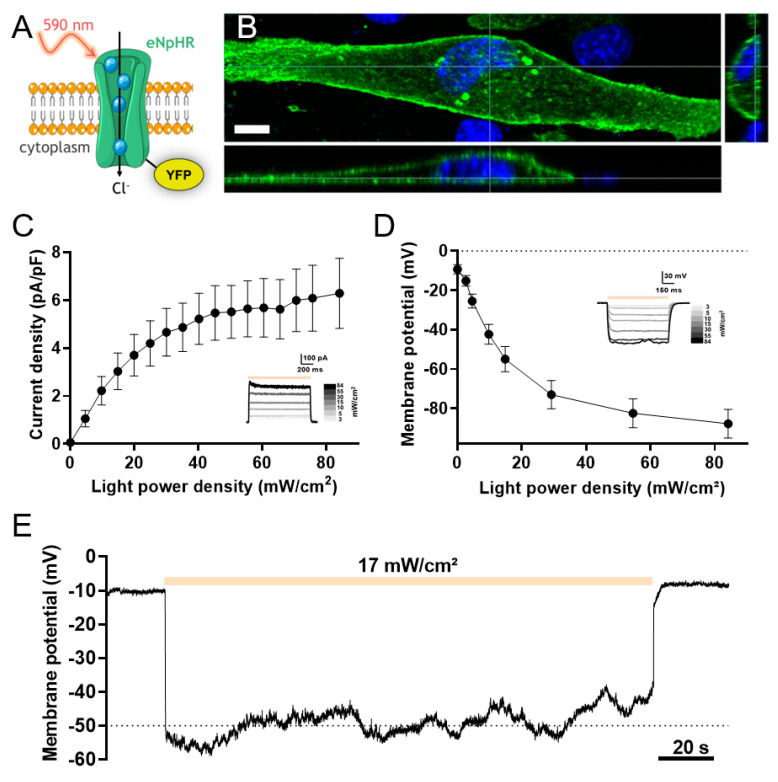
Effect of light-induced activation of the halorhopsin pump on membrane polarization of C2C12 myoblasts (**A**) Schematic representation of the light-activated chloride pump eNpHR coupled to yellow fluorescent protein (YFP). (**B**) 3D expression of eNpHR in C2C12 myoblast. YFP fluorescence highlights the cellular localization of eNpHR. Right and lower panels represent cross-sections of the myoblast (scale bar: 10 µm). (**C**) Relationship between photocurrent density and light power density. Outward eNpHR currents were recorded at a holding potential of −15 mV during a 1 s light pulse at different light intensities. The inset shows representative raw data traces recorded in response to incremental variations in light intensities (mean ± SEM, *n* = 29). (**D**) Membrane potential as a function of light power density. Membrane potentials were recorded in the current-clamp configuration (I = 0) during 1 s light pulses at different intensities. Inset shows representative traces of membrane potential modulation by light stimulation in an eNpHR-expressing myoblast (mean ± SEM, *n* = 36). (**E**) Effect of long-duration light stimulation at 17 mW/cm^2^ (orange bar) on membrane potential of an eNpHR-expressing myoblast.

**Figure 2 cells-09-01684-f002:**
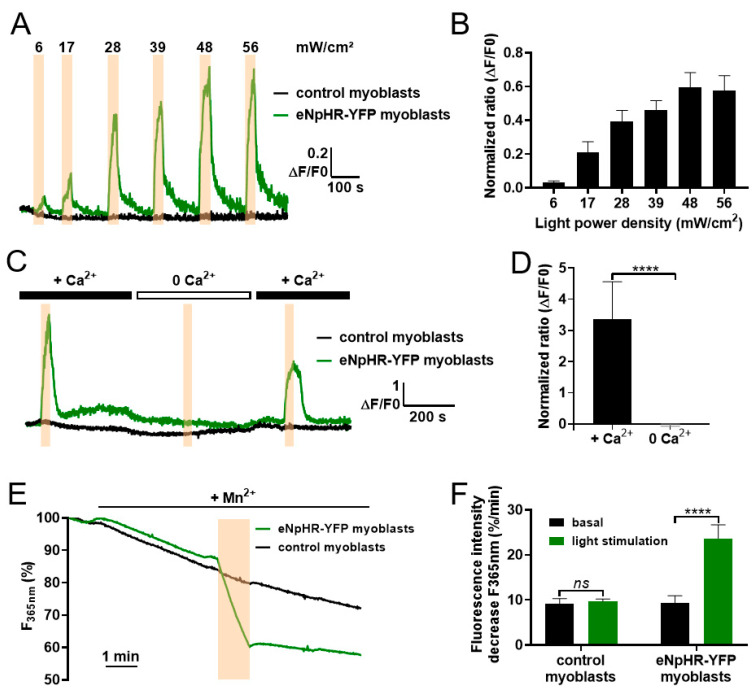
Constitutive calcium entry controlled by light stimulation of eNpHR-expressing C2C12 myoblasts. (**A**) Representative raw data traces of normalized Fura-2 fluorescence ratio at several light intensities (6, 17, 28, 39, 48, 56 mW/cm^2^, duration: 30 s, represented by orange bars) in control C2C12 myoblasts (black) and eNpHR-expressing C2C12 myoblasts (green). (**B**) Maximum amplitude (ΔF/F0) of the normalized Fura-2 fluorescence response during light stimulations of variable intensity in eNpHR-expressing C2C12 myoblasts (mean ± SEM, *n* = 31, *p* < 0.0001, Kruskal-Wallis test). (**C**) Representative traces of normalized Fura-2 fluorescence ratio during light stimulations (duration: 30 s, 48 mW/cm^2^, orange bars) in control C2C12 myoblasts (black) and eNpHR-expressing C2C12 myoblasts (green) perfused with Tyrode’s solution containing 1.8 mM Ca^2+^ or 0 mM Ca^2+^. (**D**) Amplitude (ΔF/F0) of the normalized Fura-2 fluorescence response to light stimulation of eNpHR-expressing myoblasts perfused with Tyrode’s solution containing 1.8 mM Ca^2+^ or 0 mM Ca^2+^ (mean ± SEM, *n* = 10, *p* < 0.0001, Mann–Whitney test). (**E**) Representative traces showing Mn^2+^ quenching (100 µM) of Fura-2 fluorescence recorded in control C2C12 myoblasts (gray) and eNpHR-expressing C2C12 myoblasts (green). Cells were stimulated with light (orange bar) for 1 minute at 48 mW/cm^2^. (**F**) Quantitative analysis of the maximum rate of Mn^2+^ quenching (%/min) before light stimulation (dark histograms) and during light stimulation at 48 mW/cm^2^ (green histograms) in control C2C12 myoblasts and eNpHR-expressing C2C12 myoblasts (mean ± SEM, control myoblasts n = 27; eNpHR myoblasts n = 26, *p* < 0.0001, Mann–Whitney test).

**Figure 3 cells-09-01684-f003:**
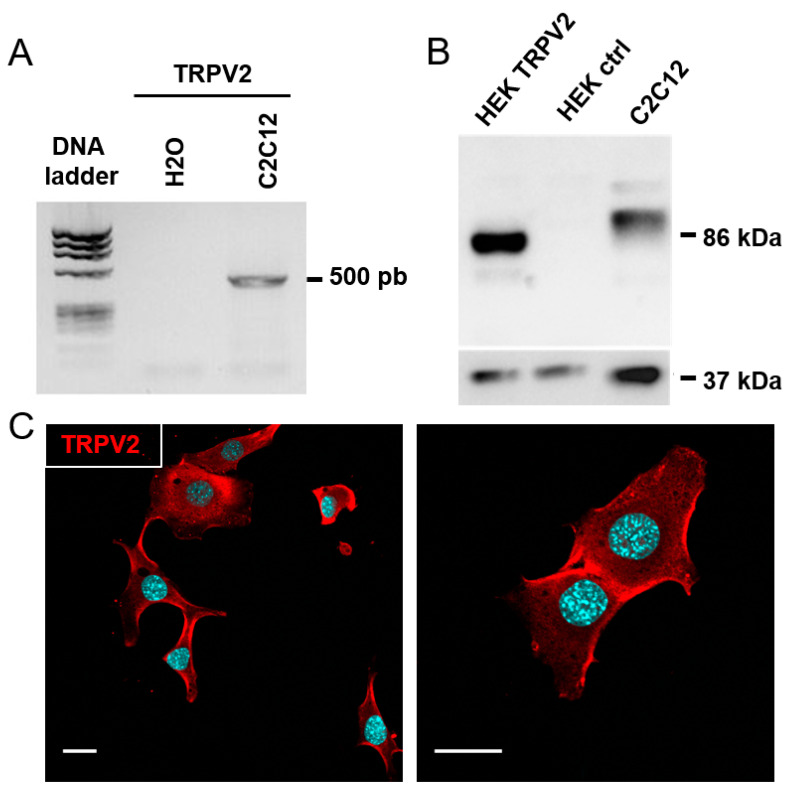
Expression of transient receptor potential vanilloid 2 (TRPV2) channels in C2C12 myoblasts. (**A**) Assessment of TRPV2 mRNA expression in C2C12 myoblasts by RT-PCR. (**B**) Western blot analysis of TRPV2 and GAPDH expression in C2C12 myoblasts, human embryonic kidney (HEK)293 cells stably expressing TRPV2 (HEK TRPV2), and control HEK293 cells (HEK ctrl). TRPV2 and GAPDH were detected sequentially on the same blot (stripped twice). 5 µg of proteins were deposited for HEK cell lysates and 10 µg for C2C12 myoblasts (*n* = 4). (**C**) Confocal images at two magnifications showing immuno-localization of TRPV2 (red) and nuclei staining (blue) in C2C12 myoblasts (scale bar: 20 µm).

**Figure 4 cells-09-01684-f004:**
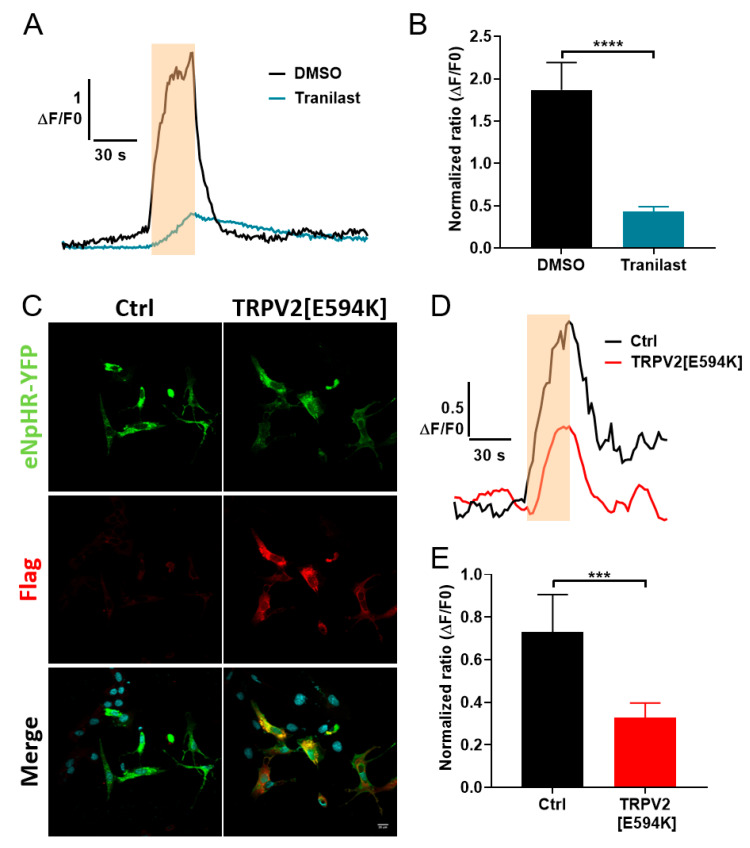
Involvement of TRPV2 channels in the mediation of constitutive calcium entry during light stimulation. (**A**) Representative traces of Fura-2 normalized ratio in response to a light stimulation of 30 s at 48 mW/cm^2^ (orange rectangle) in C2C12 myoblasts expressing eNpHR-YFP treated with DMSO 1/1000 (vehicle) or with 100 µM Tranilast, a TRPV2 inhibitor. (**B**) Maximum amplitude (ΔF/F0) of the Fura-2 fluorescence response to light stimulation in eNpHR-expressing C2C12 myoblasts treated with DMSO 1/1000 (*n* =4 0) or with 100 µM Tranilast (*n* = 45). (**C**) Confocal images showing immuno-expression of the negative-dominant TRPV2[E594K] tagged with a flag sequence (red) co-transfected (TRPV2[E594K] right panels) or not (Ctrl, left panel) with eNpHR-YFP (green) in C2C12 myoblasts. (**D**) Representative traces of normalized Fura-2 ratio during a light stimulation of 30 s at 48 mW/cm^2^ (orange rectangle) in C2C12 myoblasts expressing eNpHR-YFP co-transfected (TRPV2[E594K]) or not (Ctrl) with the negative-dominant TRPV2[E594K]. (**E**) Maximum amplitude (ΔF/F0) of the Fura-2 fluorescence response to light stimulation in control C2C12 myoblasts expressing eNpHR-YFP (Ctrl, n = 31) or in C2C12 myoblasts expressing eNpHR-YFP co-transfected with negative-dominant TRPV2[E594K] (TRPV2[E594K], *n* = 102). Data are presented as mean ± SEM. *** and **** represent significant differences with *p* < 0.001 and *p* < 0.0001, respectively (Mann–Whitney test).

**Figure 5 cells-09-01684-f005:**
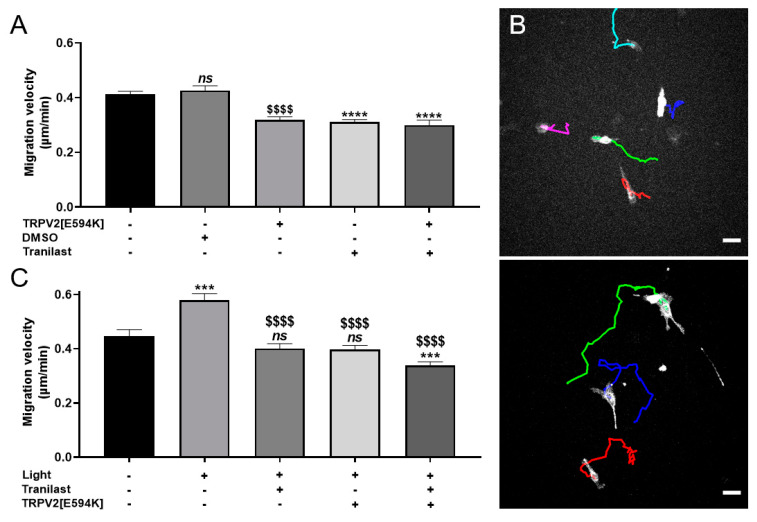
Light stimulation promotes cell migration mediated by TRPV2 in C2C12 myoblasts expressing eNpHR. (**A**) The role of TRPV2 channels in cell migration was evaluated with a cell migration tracking assay on C2C12 myoblasts expressing eNpHR. Cells were seeded at low density and the migration velocity (µm/min) was assessed for 15 h with a JuliStage system. C2C12 myoblast migration was evaluated for five conditions: without treatment (*n* = 432), co-transfected with TRPV2[E594K] (*n* = 290), treated with DMSO 1/1000 (*n* = 180), treated with 100 µM Tranilast (*n* = 285), and co-transfected with TRPV2[E594K] plus treated with 100 µM Tranilast (*n* = 89) ($ corresponds to the comparison with C2C12 myoblasts without treatment, * corresponds to the comparison with C2C12 myoblasts treated with DMSO 1/1000) ($$$$ and ****: *p* < 0.0001, Mann–Whitney test). (**B**) Cell trajectories of eNpHR-YFP-expressing C2C12 myoblasts (white cells) that were unstimulated (upper panel) or stimulated (lower panel) by light. Scale bar: 60 µm. (**C**) Effect of light stimulation on cell migration investigated by cell migration assay on C2C12 myoblasts expressing eNpHR-YFP. Cells were seeded at low density and the migration velocity (µm/min) was assessed for 15 h on C2C12 myoblasts in five conditions: unstimulated (n=115) or stimulated with light without treatment (*n* = 125), treated with 100 µM Tranilast *(n* = 133), co-transfected with TRPV2[E594K] (*n* = 162), or co-transfected with TRPV2[E594K] and treated with 100 µM Tranilast (n=163). $ corresponds to the comparison with C2C12 myoblasts stimulated by light, * and ns corresponds to the comparison with unstimulated C2C12 myoblasts. Data are shown as mean ± SEM (***: *p* < 0.001; $$$$: *p* < 0.0001, Mann–Whitney test).

**Figure 6 cells-09-01684-f006:**
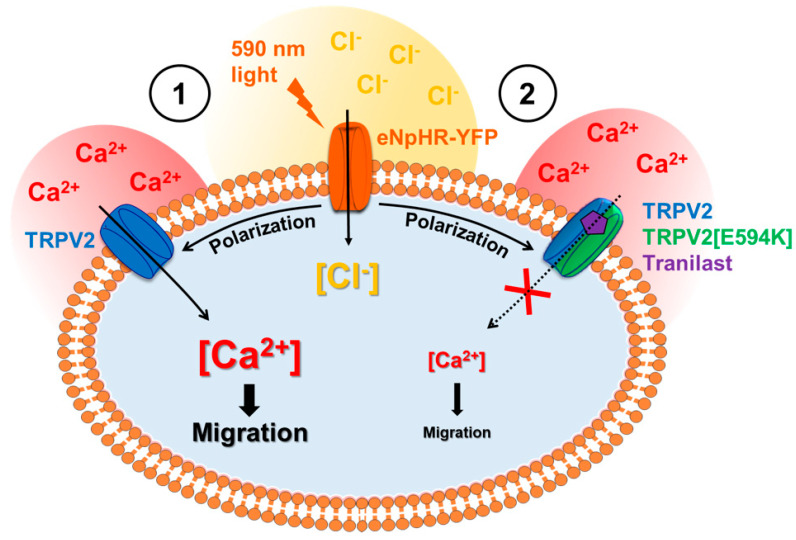
Schematic model of the effect of light-induced activation of halorhodopsin pump on the activation of calcium constitutive entry pathway through TRPV2 channels and on the modulation of cell migration. (**1**) Light stimulation of C2C12 myoblasts expressing eNpHR (orange) leads to membrane polarization by chloride ion entry (yellow) which gives rise to constitutive calcium entry (red) through TRPV2 channel (blue) by increasing the driving force for Ca^2+^ across the plasma membrane. This light-induced calcium entry increases cell migration in a manner that can be abolished (**2**) by the TRPV2 inhibitor Tranilast (purple) and the negative-dominant TRPV2[E594K] transcript (green).

**Table 1 cells-09-01684-t001:** Mouse gene-specific primers for RT-PCR.

Gene	Ref seq	Primers (5′–3′)	Annealing Temperature (°C)	Amplicon (pb)
TRPC1	NM_011643 NM_001311123	CAAGATTTTGGGAAATTTCTGG TTTATCCTCATGATTTGCTAT	55	371
TRPC4	NM 016984	TCTGCAGATATCTCTGGGAAGGATGC AAGCTTTGTTCGAGCAAATTTCCATTC	57	414
TRPM7	NM_021450 NM_001164325	TTGGAGCATTTGTGGGACAC ACGGGCTTAAATGGAGAAGCA	60	328
TRPV2	NM 011706	AGATGCTTAGAACTAAGGTGGAGG AGAGTCGGTCACGGTCAAAC	60	500
TRPV4	NM_022017	GTGGGCAAGAGCTCAGATGG CCGAGGACCAACGATCCCTAC	60	184
18S mRNA (Mrps6)	NM 080456	TTTGATTCTGAAAGCCATGCG CCAGTATGTTCTCCACAGCA	57	218

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
