# Peer review of "Fine Tuning of Calcium Constitutive Entry by Optogenetically-Controlled Membrane Polarization: Impact on Cell Migration"

_cells, 2020, doi:10.3390/cells9071684_

Round 1
Reviewer 1 Report
In this manuscript, Chapotte-Baldacci and colleagues show that TRPV2 is responsible for the calcium constitutive entry (CCE) observed in C2C12 myoblasts and that this CCE can be finely tuned using by membrane polarization controlled by a halorhodopsin chloride pump expressed at the membrane and illuminated at 590 nm
The goal of the study is well focused, the experiments are intelligently designed and the results globally convincing. The paper is well written and the figures are intelligible. Globally, the work is interesting for a large audience, well executed and the conclusions presented are supported by the data.
I have only few comments and suggestions:
- Thermodynamically, hyperpolarizing the plasma membrane by 30 mV would increase the electrochemical potential for Ca2+similarly as by increasing by 10 times the ratio of external to internal concentrations of Ca2+. So, if external [Ca2+] is increased by 10x, do you observe such influx of Ca2+ and such effect on cell mobility?
- Do you observe an effect of TRPV2 agonists such as cannabidiol ? Do you see any effect of IGF-1 that should increase TRPV2 targeting to the plasma membrane?
- What does the expression of E594K mutant do to myoblasts fusion into myotubes ?
- Is it technically feasible to illuminate the cells at only a small spot to see whether migration direction (and not only speed) can be influenced by local membrane potential?
- The progressive acquisition of membrane polarity in the process of myoblasts differentiation should be discussed. Does the technical manoeuvre described here mimick the progressive physiological hyperpolarisation observed during differentiation?
Author Response
Thank you for highlighting these points. We appreciate your kind comments, and hope that these answers will now satisfy your requirements.
- Thermodynamically, hyperpolarizing the plasma membrane by 30 mV would increase the electrochemical potential for Ca2+ similarly as by increasing by 10 times the ratio of external to internal concentrations of Ca2+. So, if external [Ca2+] is increased by 10x, do you observe such influx of Ca2+ and such effect on cell mobility?
It would be very interesting to test this hypothesis. An increase of extracellular Ca2+ by 10 times could indeed induce intracellular calcium elevation. However, we cannot rule out that a high extracellular calcium concentration could impact the function of other membrane components not involved in constitutive entries such as a reduction in the calcium extrusion by the NCX exchanger. In our opinion, the effect on intracellular calcium would therefore be less specific on calcium constitutive entries than our approach.
- Do you observe an effect of TRPV2 agonists such as cannabidiol ? Do you see any effect of IGF-1 that should increase TRPV2 targeting to the plasma membrane?
We didn’t use cannabidiol to activate TRPV2 but we tested 2-APB at 300µM and we observed a strong intracellular calcium elevation. We did not use these results because these molecules could have non-specific targets and can activate other TRP channels. The idea to test IGF-1 on TRPV2 membrane targeting and associated light induced calcium elevation is of interest. However, we have not tested this hypothesis and it would be interesting to investigate this point in future studies.
- What does the expression of E594K mutant do to myoblasts fusion into myotubes ?
Myogenesis processes involve several cellular mechanisms such as cellular proliferation, migration, fusion and differentiation. In our study, we chose to focus on myoblast migration as a model of cellular processes found in a wild range of cell types. However, it would indeed be very interesting to look at the impact of TRPV2 mutants on other myogenesis processes such as fusion and differentiation.
- Is it technically feasible to illuminate the cells at only a small spot to see whether migration direction (and not only speed) can be influenced by local membrane potential?
Yes, it is possible to stimulate a small spot in a cell by using laser illumination modules allowing to target a subcellular area. We discussed this possibility in the paper based on an article from the laboratory (Sebille et al,. 2017) where we used a FRAP module to stimulate very locally the channelrhodopsin 2, a cationic channel activated by blue light. Moreover, new commercially available devices such as the Mightex system (https://www.mightexbio.com/products/patterned-illumination/polygon/), also allow subcellular illumination. This approach could be of particular interest in cellular models such as migrating fibroblasts that display very large lamellipods.
- The progressive acquisition of membrane polarity in the process of myoblasts differentiation should be discussed. Does the technical manoeuvre described here mimick the progressive physiological hyperpolarisation observed during differentiation?
Membrane polarization is observed in myoblasts during the fusion process. We think that this technique could be used to reproduce the progressive membrane polarization observed in physiological condition by using for example, a slow ramp of light. This point is now added to the discussion (line 474).
Reviewer 2 Report
This is a carefully controlled and interesting study that demonstrates a role for constitutive calcium entry in driving cell migration. The authors adopt an optogenetic approach and show that 590 nm light exposure activates halorhodopsin, resulting in a light-intensity-dependent hyperpolarization and increased calcium entry. A pharmacological tool along with expression of a dominant negative channel both suggest a major role for TRPV2. Finally, the authors clearly demonstrate a role for constitutive calcium entry in enhancing cell migration. Overall, this is a very nice study. I do have a few relatively minor comments.
- Can the authors quantify how quickly hyperpolarization occurs, particularly the time constant to reach a new membrane potential level? It looks complete within <5 s from Figure 1 but it would be helpful to know more precisely, particularly if a researcher wants to express a rapidly inactivating ion channel.
- Following the hyperpolarization, can the authors estimate the change in intracellular chloride concentration? This could be important because chloride can regulate multiple signalling pathways.
- After the first light pulse in Figure 2C, there is a secondary rise in cytosolic calcium. This is typical of calcium-induced calcium release. Do the authors think calcium entry could trigger CICR? Also, in Figure 2C, why is the 3rd light pulse (2nd in presence of calcium) so much less effective than the first? Could this be because the store have not refilled for CICR?
- A major source of constitutive calcium entry is through Orai1 channels. The authors mention this but focus on TRPV2. Is there any role for Orai1 here?
Author Response
Thank you for highlighting these points. We appreciate your kind comments, and hope that these answers will now satisfy your requirements.
- Can the authors quantify how quickly hyperpolarization occurs, particularly the time constant to reach a new membrane potential level? It looks complete within <5 s from Figure 1 but it would be helpful to know more precisely, particularly if a researcher wants to express a rapidly inactivating ion channel.
We performed new analysis of patch clamp data and found a time constant of 18.7 ± 2.1 ms to reach the new membrane potential at maximum light intensity. This value is close to previously published values by Zhang et al., 2007 (35.6 ±- 15.1). This is now specified in the results section of the paper (line 262).
- Following the hyperpolarization, can the authors estimate the change in intracellular chloride concentration? This could be important because chloride can regulate multiple signalling pathways.
Thank you for this comment. Indeed, halorhodopsin activation probably increases intracellular chloride concentration unfortunately we didn’t evaluate this concentration. Chloride ions play a main role in the control of the cell membrane potential but, to our knowledge, there is no direct chloride concentration dependent calcium signaling pathways.
- After the first light pulse in Figure 2C, there is a secondary rise in cytosolic calcium. This is typical of calcium-induced calcium release. Do the authors think calcium entry could trigger CICR? Also, in Figure 2C, why is the 3rd light pulse (2nd in presence of calcium) so much less effective than the first? Could this be because the store have not refilled for CICR?
Indeed, this point is very interesting. Unfortunately, we did not investigate CICR processes in our model and it would be interesting to do it. We cannot rule out that CICR takes part in our calcium response and further investigation are needed to answer this question. Potential involvement of CICR in the calcium response is now evoked in the discussion (line 426).
- A major source of constitutive calcium entry is through Orai1 channels. The authors mention this but focus on TRPV2. Is there any role for Orai1 here?
Indeed, ORAI could participate in calcium entries. TRPV2 inhibition with Tranilast reduced significantly light-induced calcium entry which suggests its strong contribution to these entries. However, the remaining calcium elevation could indeed be due to other TRP and Orai channels. This is now added to the discussion (line 419 and line 426)
Reviewer 3 Report
In this manuscript, the authors demonstrate a useful optonegetically technique to control membrane potential in native cells. They showed that halorhodopsin can achieve a fine-tuned control of membrane potential in C2C12 cells. In addition, Ca2+entry though TRPV2, which was regulated by halorhodopsin-dependent membrane polarization, modulated the cell-migration. Overall, experimental results seem to be solid and the manuscript is well written. This reviewer concerns following two points.
- Judging from methods of electrophysiology, the authors used roughly symmetrical Cl- concentration in the bathing (~150 mM) and pipette (141 mM) solution. This makes the equilibrium potential of Cl- around 0 mV. On the other hand, the activation of halorhodopsin by light hyperpolarized C2C12 cell to -87.8 mV (line260, p8). Even intracellular Cl- concentration is quite high under this experimental condition, how can this pump produce Cl- entry? Is this pump electromotive? These explanations should be included.
- In Fig.2 and 4, why did the authors use F/F0 as a Fura-2 signal indicator? The ratio is more usual and useful for experiments using Fura-2. Indeed, when the authors use the ratio, they can also analyze the change of basal Ca2+ level between wild and mutant-TRPV2 expressing C2C12 cells (Fig4C-4E). This can explain the significant difference of migration between control and mutant-TRPV2 expressing cells in Fig.5A.
Author Response
Thank you for highlighting these points. We appreciate your kind comments, and hope that these answers will now satisfy your requirements.
- Judging from methods of electrophysiology, the authors used roughly symmetrical Cl- concentration in the bathing (~150 mM) and pipette (141 mM) solution. This makes the equilibrium potential of Cl- around 0 mV. On the other hand, the activation of halorhodopsin by light hyperpolarized C2C12 cell to -87.8 mV (line260, p8). Even intracellular Cl- concentration is quite high under this experimental condition, how can this pump produce Cl- entry? Is this pump electromotive? These explanations should be included.
Halorhodopsin pump is light activated. In halorhodopsin, absorption of light by the covalently bound retinal chromophore triggers a reaction cycle during which the protein passes through a sequence of 6 intermediate states, of which the last two steps are associated with, respectively, the release and uptake of a chloride ion even against the chloride concentration gradient (Chow et al., 2012 (PMID: 22341320); Feroz et al., 2018 (PMID: 30021110) and 25 of the manuscript). As a consequence, this pump can induce chloride entry even against its electrochemical gradient. This point is now specified in the manuscript (line 78).
- In Fig.2 and 4, why did the authors use F/F0 as a Fura-2 signal indicator? The ratio is more usual and useful for experiments using Fura-2. Indeed, when the authors use the ratio, they can also analyze the change of basal Ca2+ level between wild and mutant-TRPV2 expressing C2C12 cells (Fig4C-4E). This can explain the significant difference of migration between control and mutant-TRPV2 expressing cells in Fig.5A.
Indeed, it would have been interesting to compare basal calcium levels using the ratio F340/F380. Unfortunately, we encountered a light illumination device issue during the experiments leading us to change this device. As a consequence, we thought it would be not reasonable to compare the ratio between before and after device replacement and we therefore chose to only compare F/F0. Nevertheless, for this reviewing process, we tried to analyze the data and observed no difference in the calcium basal level between the control and the mutant. This suggests that the difference observed in cell migration is not related to a different basal calcium level recorded at resting membrane potential. A hypothesis would be that the difference between control and mutant can be observed only in more polarized potentials as observed with light stimulation. As mentioned in our introduction, several studies have shown that non-excitable cells exhibit membrane potential variations characterized by hyperpolarization phases during essential cellular processes such as proliferation, migration or differentiation (references 7-9 of the manuscript). Such a mechanism could therefore explain the difference in migration observed between control and mutant-TRPV2 expressing cells.
Reviewer 4 Report
The authors showed that acrivating halorhodopsin chloride channels in the C2C12 cells hyperpolarize the cells increasing [ca2+]i. They also showed that this rise in ca2+ is mediated by TRPV2 activation. Although there are some novelties in the results, the data are preliminary and need more thorough experiments. It is not obvious to me how chloride channels activate TRPV2 as well as how TRPV2 activation increases cellular migration.
The authors need to repeat the experiments conducted in FIg 1 in non-transfected cells to show the effect of light on normal cells.
As the authors mentioned the effect of cl- channel activation on [ca2+]i is not direct; then they need to explain why the rise in [ca2+]i was observed immediately after light application.
It needs to be explained why the rise in [ca2+]i stayed up for a long time but the changes in membrane potential was immediately returned to original values after shutting down the light.
The authors need to mention P values wherever they talk about significant differences such as line 288 as well as the test applied.
For fig 3C and to claim that TRPV2 is located in the plasma membrane, the authors need to stain the plasma membrane and do colocalization analysis.
Author Response
Thank you for highlighting these points. We appreciate your kind comments, and hope that these answers will now satisfy your requirements.
- The authors need to repeat the experiments conducted in FIg 1 in non-transfected cells to show the effect of light on normal cells.
Halorhodopsin is now widely used in the literature and such an effect of light on untransfected cells has never been reported. Besides, in our calcium experiments, the light was shed on a large population of cells within which both transfected and untransfected cells were present. As mentioned in the manuscript line 278, none of the untransfected cells responded to light stimulation. This discard in our opinion a potential side effect of light.
- As the authors mentioned the effect of cl- channel activation on [ca2+]i is not direct; then they need to explain why the rise in [ca2+]i was observed immediately after light application.
The time constant for membrane polarization obtained at maximum light intensity has been calculated and now added to the results and is of 18.7 ± 2.1 ms (n=36). According to our hypothesis, the membrane polarity drives calcium entries, it is therefore not surprising that calcium elevation is observed immediately after a light application.
- It needs to be explained why the rise in [ca2+]i stayed up for a long time but the changes in membrane potential was immediately returned to original values after shutting down the light.
We believe that kinetics of calcium rising and decrease are different because they involve different mechanisms. Indeed, calcium elevation is the consequence of constitutive entries drove by membrane polarization. Moreover, it is not excluded that a Calcium-Induced Calcium Release (CIRC) mechanism also takes part in light-induced calcium elevation. This point is now added to the discution (line 426). Besides, these stimulations led to a large calcium increase that necessitates a strong calcium extrusion through actors SERCA and PMCA. This extrusion process is slower than the membrane potential variation obtained when the light was turned off and is not dependent on it (PCMCA and SERCA). This could explain the difference you mentioned in your comment.
- The authors need to mention P values wherever they talk about significant differences such as line 288 as well as the test applied.
Both P values and statistical tests are systematically specified in the figure legends.
- For fig 3C and to claim that TRPV2 is located in the plasma membrane, the authors need to stain the plasma membrane and do colocalization analysis.
In addition to our immunocytochemistry experiments we performed pharmacological tests as well as the expression of the TRPV2 mutant that clearly show the functionality of TRPV2. In the overall, the combination of all these experiments is sufficient in our opinion to demonstrate that TRPV2 is functional at the plasma membrane.
Round 2
Reviewer 4 Report
It has improved.